# Data privacy during pandemics: a systematic literature review of COVID-19 smartphone applications



Amany Alshawi[1], Muna Al-Razgan[2], Fatima H. AlKallas[2], Raghad Abdullah Bin Suhaim[2], Reem Al-Tamimi[2], Norah Alharbi[2] and Sarah Omar AlSaif[2]

[1] King Abdulaziz City for Science and Technology (KACST), Riyadh, Saudi Arabia
[2] King Saud University (KSU), Riyad, Saudi Arabia

## ABSTRACT

**Background:** On January 8, 2020, the Centers for Disease Control and Prevention officially announced a new virus in Wuhan, China. The first novel coronavirus (COVID-19) case was discovered on December 1, 2019, implying that the disease was spreading quietly and quickly in the community before reaching the rest of the world. To deal with the virus' wide spread, countries have deployed contact tracing mobile applications to control viral transmission. Such applications collect users' information and inform them if they were in contact with an individual diagnosed with COVID-19. However, these applications might have affected human rights by breaching users' privacy.

**Methodology:** This systematic literature review followed a comprehensive methodology to highlight current research discussing such privacy issues. First, it used a search strategy to obtain 808 relevant papers published in 2020 from well-established digital libraries. Second, inclusion/exclusion criteria and the snowballing technique were applied to produce more comprehensive results. Finally, by the application of a quality assessment procedure, 40 studies were chosen.

**Results:** This review highlights privacy issues, discusses centralized and decentralized models and the different technologies affecting users' privacy, and identifies solutions to improve data privacy from three perspectives: public, law, and health considerations.

**Conclusions:** Governments need to address the privacy issues related to contact tracing apps. This can be done through enforcing special policies to guarantee users privacy. Additionally, it is important to be transparent and let users know what data is being collected and how it is being used.

Corresponding author
Amany Alshawi,
aalshawi@kacst.edu.sa

## INTRODUCTION

At the end of December 2019, a new COVID-19 virus appeared in Wuhan, China. The novel virus severe acute respiratory syndrome (SARS-CoV-2), a COVID-19 virus family member, produces an infectious disease known as COVID-19, causing illnesses that vary from the common cold to more severe diseases (*Sharma et al., 2020*; *Skoll, Miller & Saxon, 2020*).

The rapid global spread of COVID-19 virus overwhelmed health sectors and caused a significant worldwide public health crisis, prompting the World Health Organization (WHO) to declare COVID-19 a global pandemic (*Whaiduzzaman et al., 2020*). The virus has affected human life in many different aspects: travelling, education, business, entertainment, and others (*Mbunge, 2020*).

Due to the virus' spread, many countries have taken measures and imposed restrictions like lockdowns, limiting activities requiring human gathering and interactions (*De, Pandey & Pal, 2020*). Consequently, national governments seek solutions to minimize infected cases, which results in employing digital surveillance technologies to contain the virus. A few countries have managed and controlled the pandemic using such technologies and developing smartphone apps and information system technologies to control the virus' spread (*Dwivedi et al., 2020*). Specifically, when an individual is diagnosed with COVID-19, anyone close to them during the contagious period must quarantine for 2 weeks (*Cho, Ippolito & Yu, 2020*). This solution requires individuals to download the app and access their sensitive data; therefore, it raises significant privacy issues.

Contact-tracing apps (*Vitak & Zimmer, 2020*) collect a wide range of data, including personal, location, and health and fitness data. In many countries, individuals avoid using these apps due to privacy concerns. Debates about the COVID-19 app's ethics have been largely preoccupied with privacy concerns, as these data is shared with governments, health ministries, and organizations (*Hendl, Chung & Wild, 2020*). It is an individual's right to understand what data these apps have access to, who has privileges to obtain them, and how their data is used (*Vitak & Zimmer, 2020*). Digital solutions must comply with confidentiality requirements (privacy and security) to ensure personal data protection. Examples of these actions could be obtaining users' consent, transparency, voluntary self-reporting, and anonymization. Moreover, clear and ethical principles must be stated (*Hendl, Chung & Wild, 2020*). When designing and deploying these apps and any other solutions, it is necessary mitigating privacy concerns.

There are a few literature reviews focused on apps developed for contact-tracing, prevention, surveillance measures, and mapping disease spread. In *Jalabneh et al. (2020)* were identified 17 primary studies whose current application is to monitor and diagnose infected individuals. However, the authors only share their view concerning data privacy regarding how users' information is not necessarily accurate, which affects data analysis. *Golinelli et al. (2020)* identified 52 articles classified into seven categories covering public health needs addressed by recognized digital solutions in the COVID-19 pandemic context. The authors address privacy in two of their identified categories, mentioning only that it would preferable the applications do not collect personal data (*Golinelli et al., 2020*; *Verma & Mishra, 2020*) provides a systematic review of smartphone technology applied in the fight against COVID-19.

Regarding the collection of users' information, there is only mention of an app developed in the United Kingdom, where the data are stored in various places. *Zimmermann et al. (2021)* provided a set of population perceptions about contact-tracing apps regarding authority trusting and individual privacy, among others. However, the context of such work only considers the three main German-speaking countries: Germany,

Austria, and Switzerland. Finally, *Hussein et al. (2020)* provides a review of several digital health surveillance systems where regulations and data protection are only approached from the perspective of the pressure on users to share their personal information to access such systems and apps.

As evident, several studies have been conducted focusing on contact-tracing apps developed during the COVID-19 pandemic. Some of those studies discussed such apps about privacy and ethical concerns. Nevertheless, there is an important window of opportunity for how such information technologies can help health ministries and other parties contain the virus' spread. Our overarching goal is to provide a better study of privacy concerns in the context of COVID-19 apps. Toward this goal, we examined and analyzed the existing studies on COVID-19 apps and privacy concerns and their findings, and summarized this research's efforts.

The remainder of this paper is as follows: "Survey Methodology" introduces our methodology for paper selection and data collection, "Quality Assessment Criteria" represents our collected studies' analysis, "Discussion" highlights implications, "Limitations and Future Work" discusses future work and limitations, and "Conclusion" concludes.

## SURVEY METHODOLOGY

To ensure the accuracy of our systematic literary review results, we adapted and modified Liao's methodology, proposed in *Liao et al. (2020)*. The methodology goes through six stages as shown in Fig. 1.

### Research questions

We aim to answer, at the end of this systematic literary review, three research questions:

1. What techniques are proposed to protect users' privacy in digital surveillance?
2. How does the law protect users' privacy in COVID-19 applications?
3. How do different entities contribute to preserving individuals' health privacy?

### Search strategy

The search strategy consists of three steps, as defined below.

### *Finding keywords*

To find keywords related to our topic, we used Nails Project (*Knutas et al., 2015*; *Salminen, Knutas & Hajikhani, 2020*). We then chose the words that we deemed to be the most relevant to the systematic literary review topic and those most accurate from among the candidate words.

### *Forming search strings*

From the selected words, we formalized two strings to use in the search process:

2.1 "Privacy" AND ("mobile application" OR "apps") AND ("COVID-19" OR "COVID-19" OR "COVID-19 virus" OR "COVID-19 pandemic").

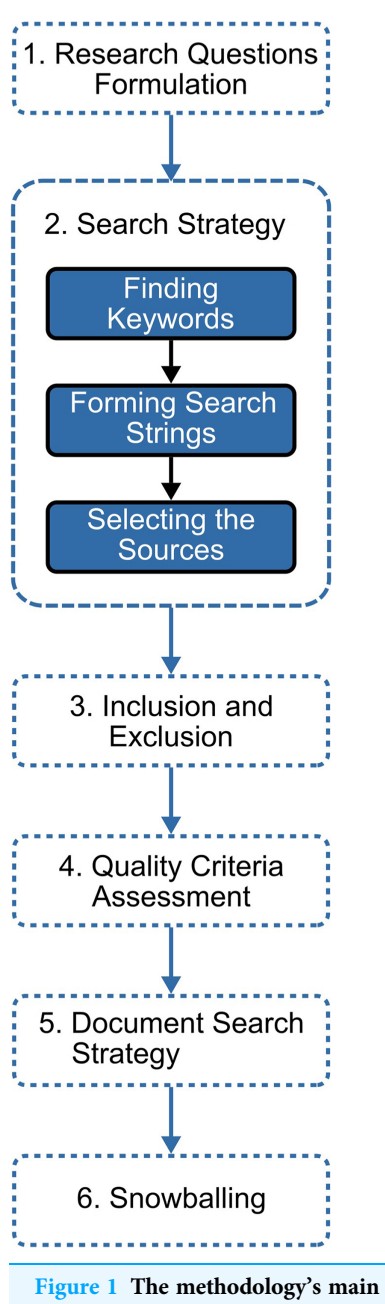

**Figure 1  The methodology's main stages.**     

2.2 "Privacy" AND ("mobile application" OR "apps") AND ("COVID-19" OR "COVID-19" OR "COVID-19virus" OR "COVID-19 pandemic") AND ("contact-tracing" OR "location privacy" OR "data protection" OR "privacy protection").

This systematic literary review focuses on three subjects: "COVID-19" and "privacy" in "mobile applications". To ensure coverage of all papers across the field, we chose synonyms for each term. In the first string, the results will be general, including any papers mentioning the three main subjects. In the second string, we narrowed the results to papers related to "contact-tracing" and combined it with keywords related to this term.

**Table 1 Inclusion and exclusion criteria.**

| Inclusion criteria | Exclusion criteria |
|---|---|
| English language | Non-English language |
| Between 2019 and 2020 | Before 2019 |
| Journals and magazines | Any document types other than journals and magazines |
| Use exact keyword strings | Papers that mention COVID-19 without privacy, and vice versa |

**Table 2 Quality assessment questions.**

| Q.ID | Quality assessment questions |
|---|---|
| Q1 | Is the aim clearly stated? |
| Q2 | Have any COVID-19 privacy issues been reported? |
| Q3 | Has it answered its RQs? |
| Q4 | What are the applied techniques related to COVID-19? |

### Selecting sources

For the research process, we selected 10 libraries. They are the most reliable and provide the highest quality of research: Microsoft Academic (https://academic.microsoft.com/home), Wiley Online Library (https://onlinelibrary.wiley.com/), IEEE Xplore (https://ieeexplore.ieee.org/Xplore/home.jsp), Sage Journals (http://us.sagepub.com), Taylor & Francis Group (https://taylorandfrancis.com/online/), Springer Link (https://link.springer.com/), Science Direct (https://www.sciencedirect.com/), Scopus Preview (https://www.scopus.com/home.uri), ACM Digital Library (https://dl.acm.org/), and Web of Science (https://www.webofknowledge.com/).

## Inclusion and exclusion criteria

After exploring all the libraries' engines, we obtained 808 papers, which decreased to 565 papers after duplication removal. We then filtered the papers by reading titles, keywords, and abstracts. This process yielded 60 papers. We used the inclusion and exclusion criteria to filter the papers as shown in Table 1.

## Quality assessment criteria

The 60 papers from the previous step were read thoroughly and checked against quality assessment questions as in Table 2. All questions were weighted to 1 point for Yes, 0 for No, and 0.5 for partial. Papers that scored 2 points or more were included in the final collection as shown in Fig. 2. Papers scoring less than 2 points were reviewed by a team member and re-scored. Papers still scoring less than 2 were excluded.

## Document search strategy

After assembling the final collection of papers, we assigned each paper to its origin library. As shown in Table 3, some libraries have no papers in the final collection.
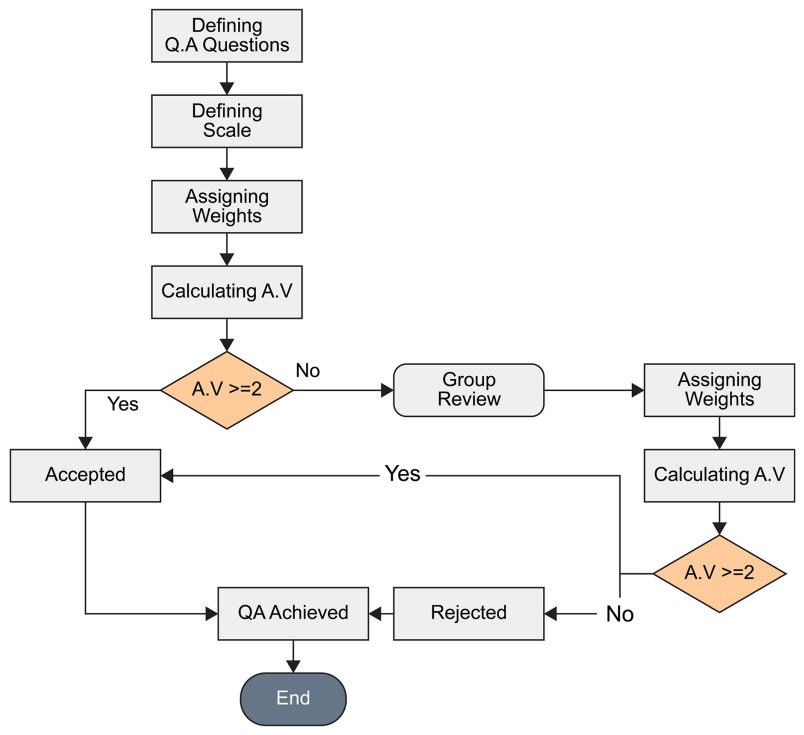

**Figure 2 Quality assessment flow chart.**   

**Table 3 Included and excluded papers.**

| Data source | Search results | Candidate papers | Primary studies |
| --- | --- | --- | --- |
| Microsoft Academic | 142 | 4 | 3 |
| Wiley | 157 | 10 | 3 |
| IEEE | 18 | 8 | 6 |
| Sage Journals | 134 | 12 | 7 |
| Taylor & Francis | 113 | 6 | 3 |
| Springer Link | 14 | 3 | 2 |
| Science Direct | 97 | 6 | 6 |
| Scopus | 58 | 10 | 5 |
| ACM | 41 | 0 | 0 |
| Web of Science | 34 | 1 | 0 |
| Total | 808 | 60 | 35 |

## Snowballing

For better comprehensiveness of papers, we conducted forward snowballing to cover all the papers related to our topic. After reading the titles of the papers' references, we obtained 13 papers in the first stage. After reading the abstracts, the number decreased to 6 papers. We then read the papers in their entirety and performed quality assessment, yielding five papers. This process was repeated until no more results were produced;

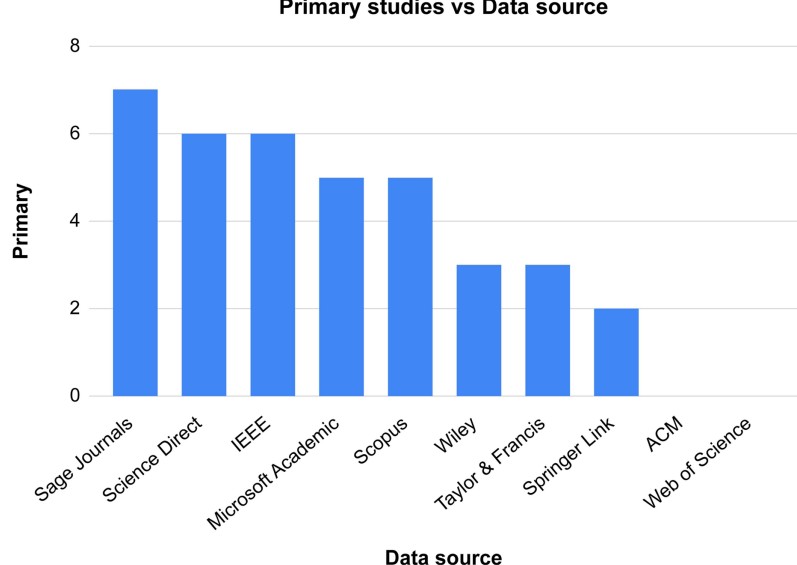

**Figure 3** Total number of papers from each data source.

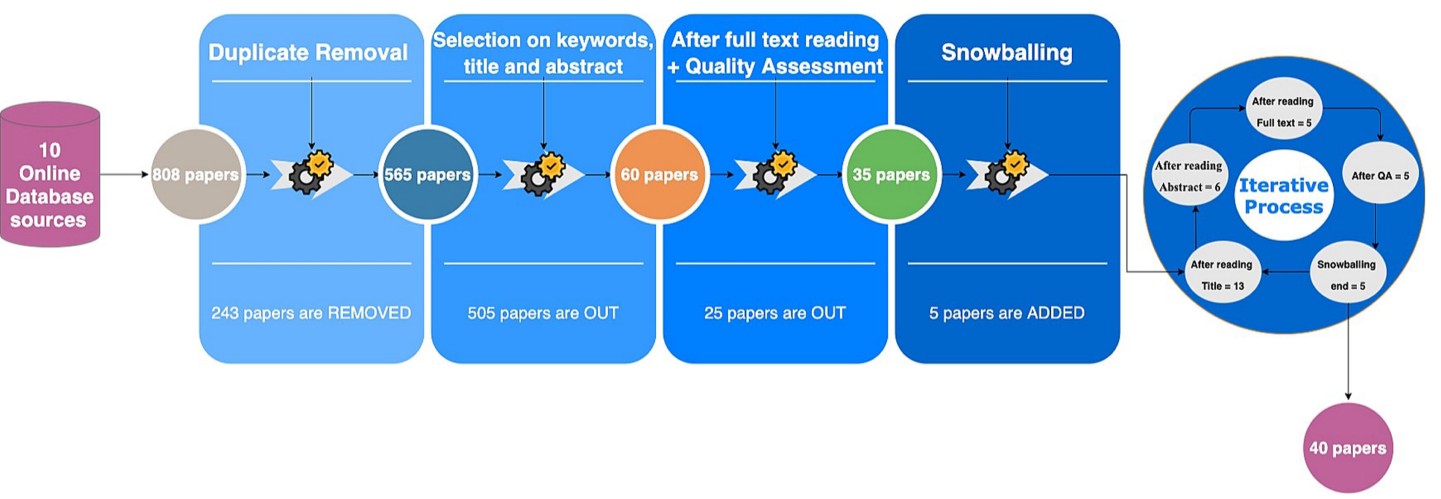

**Figure 4** Paper selection process.

resulting in 40 papers. Figure 3 shows the total number of papers from each data source and Fig. 4 illustrates the selection process.

## DISCUSSION

In January 2020, WHO declared the newly-identified COVID-19 virus as a global pandemic (*Whaiduzzaman et al., 2020*). When a COVID-19 patient has face-to-face contact with another person for 15 min or more and the distance between them is less than 1.5 m, there is a high possibility the other person will become infected with the virus (*Garg et al., 2020*). Therefore, when an individual is in close contact with a person

diagnosed with COVID-19, the individual is advised to quarantine themselves for approximately 2 weeks (*Cho, Ippolito & Yu, 2020*).

This process was easy to manage at the pandemic's beginning. However, with the virus' widespread, it became difficult and time-consuming (*Whaiduzzaman et al., 2020*), necessitating the tracing of infected, suspected, and contact persons in relation to COVID-19 patients.

Unlike vaccines, requiring time for development and approval (*Joo & Shin, 2020*), population-wide contact-tracing applications can more immediately control viral spread and enable the successful containment of COVID-19 or any future infectious disease (*Skoll, Miller & Saxon, 2020*; *Dwivedi et al., 2020*; *Riemer et al., 2020*). Many COVID-19-affected countries look at these technology-based solutions, facilitating and automating limiting infection and minimizing viral spread. These can be deployed following different approaches and adapting multiple technologies, such as a global positioning system (GPS), Wireless Fidelity (Wi-Fi) technology, and Bluetooth (*Mbunge, 2020*).

Contact-tracing refers to identifying an individual and their contacts (*Vitak & Zimmer, 2020*). In addition to administering infected cases, contact-tracing apps trace the infection route from the diagnosed individual to those with whom they have been in close contact. Traditional contact-tracing is a strategy proposed more than 80 years ago. It was used as a part of the response to any disease outbreak and has been implemented to control infectious diseases like the severe acute respiratory syndrome (SARS) epidemic, since it is easy to adopt at any time (*McLachlan et al., 2020*; *Fahey & Hino, 2020*; *Trang et al., 2020*).

Different approaches exist to develop contact-tracing. The first order app identifies only the individuals in direct contact with the patient. The single-step app was the enhanced version of the first step. It identifies any individual in contact with an infected patient and any who became infected, along with their contacts, and so on. Different apps, such as iterative and retrospective, have many limitations and have failed to achieve their purpose. To bridge the gap, modern contact-tracing apps were proposed, which rely on technologies like wireless and Bluetooth. These apps have many features: live maps of confirmed cases, location-based tracking, and quarantine and isolation monitoring. Even with these applications, from a public health viewpoint, this tactic might not be ethically effective (*Rowe, 2020*).

Countries worldwide have taken different approaches and applied different technologies and models to develop and roll out contact-tracing applications. Table 4 summarizes contact tracing applications developed in different countries during the COVID-19 pandemic along with their privacy concerns. A distribution of contact-tracing applications around the world is illustrated in Fig. 5.

Contact-tracing apps collect sensitive personal data like phone numbers, MAC addresses, and GPS location data (*McLachlan et al., 2020*; *Cao et al., 2020*). Individuals' perceptions of these applications vary; however, users' primary concern while using these apps is privacy, which is the key factor motivating many to refrain from downloading them (*Sharma et al., 2020*; *Goggin, 2020*; *O'Leary, 2020*).

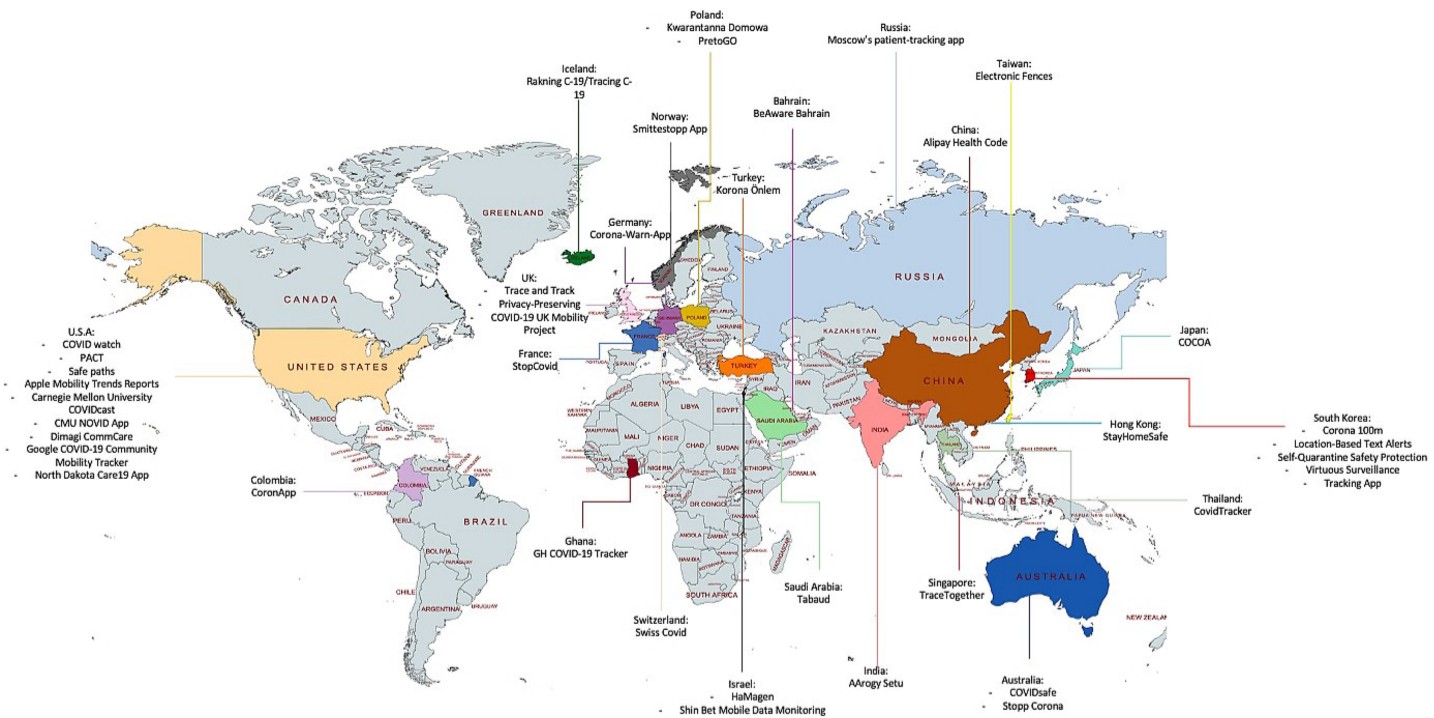

**Figure 5** Map to visualize contact-tracing applications around the world. 

Contact-tracing raises significant privacy concerns and questions about user privacy (*Whaiduzzaman et al., 2020*; *Cao et al., 2020*). Some considerations to bear in mind are: Are individuals willing to share their contacts and locations with governments and health authorities? What will happen to these data once the pandemic ends? What is this data's lifespan? For what purpose will this data be used? (*Dwivedi et al., 2020*; *Vitak & Zimmer, 2020*; *O'Leary, 2020*). Privacy risks can be mitigated by obtaining consent and safeguarding individuals' privacy by giving them control over how their data are collected and used to encourage the continued voluntary use of these apps (*Mbunge, 2020*; *Trang et al., 2020*; *Nanni et al., 2020*; *Wang & Liu, 2020*; *Klar & Lanzerath, 2020*).

Countries worldwide have developed and deployed contact-tracing apps and adapted different approaches to mitigate privacy concerns (*Wang & Liu, 2020*). Two technology-based solutions, namely Bluetooth and GPS, are used to implement these apps. Bluetooth-based apps identify whether two individuals are in the same place and whether there is a distance of at least 1.5 m between them. It does not collect exact location information, but, ironically, is more accurate. Users may feel they have a greater degree of privacy and may be less concerned about being monitored 24/7. In contrast, GPS-based apps collect individuals' data on a 24/7 basis.

## Proposed techniques to protect users' privacy in digital surveillance

Based on how applications collect and share information, app architecture is classified into two different models: centralized and decentralized. These models differ in their

**Table 4 Contact-tracing applications around the world along with their privacy concerns.**

| Application name | Country | Model | Technology | Privacy concerns |
|---|---|---|---|---|
| AArogy Setu | India | Centralized | GPS+ Bluetooth [26] | This app is mandatory in India to reduce COVID-19 cases, app privacy policy has been deleted from apple and google stores and individual data is hashed and stored as an anonymous on government server. |
| Apple Mobility Trends Reports | US | NA | NA | This app provides the policies in Apple that specifies data is encrypted, but there is no mention to the data deletion process. |
| BeAware Bahrain App | Bahrain | NA | GPS | The data is encrypted, and the users have the ability to request deleting their data. Also, it has access to users' phone numbers which raise privacy concerns. |
| Carnegie Mellon University COVIDcast | US | NA | NA | This app does not have clear policies provided with it, and once individual files the survey there is no deletion on data and data encrypted not specified. |
| CMU NOVID App | US | Decentralized | Bluetooth | This app provides NOVID policies that mention each user gets a random anonymous ID and notification token. It encrypts the individual ID, and individuals can request deletion of data and copy of data at any time, also data may be shared with other companies anonymously. |
| COCOA | Japan | NA | Bluetooth (Google/ Apple API) | This app faces privacy concerns upon the third-party API's used since it may reach the data, another concern is that this app does not provide control over data. |
| COVID-19 100m | South Korea | Centralized | GPS | Collects diagnose data, nationality, gender, age and locations. The individual locations only are tracked and stored to notify them if there was an infected individual visiting the same location. |
| COVID-19-Warn-App | Germany | Decentralized | Bluetooth (Google/ Apple API) | Personal data is not stored which make the individual trust the use of it, but it has two concerns the first one there is lack of data control from user, the second concern is using API as a third party raise the privacy concern of personal information access from Google and Apple API's. |
| COVID-19pp | Colombia | NA | GPS | Collects users' location data |
| COVID-19virus Impact Dashboard | Latin America | NA | NA | Regarding privacy, this app does not provide any privacy policies which raise privacy concerns, since it does not encrypt the data and it does not provide any information about data deletion. |
| COVID watch | U.S.A | Decentralized | Bluetooth | Used by volunteers from many different countries to exposure alerts when some infected users become near. Papers did not discuss its privacy violence clearly but for sure it has been mentioned as one of the applications that has privacy problems. |
| COVID-19 UK Mobility Project | UK | Decentralized | NA | This app accesses the device ID and personal information, these data may be deleted by the user, but data encryption is not mentioned on the privacy policies provided. |

| Application name | Country | Model | Technology | Privacy concerns |
|---|---|---|---|---|
| COVIDsafe | Australia | Centralized | Bluetooth (BlueTrace protocol) | This app provides a level of privacy since it provides the user the freedom to delete the data anytime which will delete the entire application data from the phone, but these data are stored on the government server unencrypted and it will be deleted at the end of the pandemic. There is a privacy concern since the data on the government server is not encrypted which makes it open for the attackers to steal it. |
| CovidTracker | Thailand | Centralized | GPS | It collects both infected and healthy users' information and sends it to a centralized server that is controlled by government and authority health. |
| Dimagi CommCare | U.S.A | NA | NA | This app does not provide data encryption, so it raises the privacy concern since it stores personal data on the server, but it allows the user to request deletion of the data. |
| Electronic Fences | Taiwan | Centralized | GPS | Collects the location and insurance card data and sends it to the Central Epidemic Command Center (CECC) to check their last 14-days activity. |
| GH COVID-19 Tracker | Ghana | NA | GPS | Collects users' location data. Papers did not discuss its privacy violence clearly but for sure it has been mentioned as one of the applications that has privacy problems. |
| Google COVID-19 Community Mobility Tracker | U.S.A | Centralized | GPS | This app's privacy policy is controlled by Google, location data is stored locally by Google, and the encryption of data is done on the transit of data to the server. |
| HaMagen | Israel | Decentralized | GPS+Bluetooth | This app does not have privacy risk since it provides the policies to all the individuals, and it stores personal data on the individual device unless this individual is infected his/her data will be known. When the individual deletes the application, all of the data will be deleted. |
| Alipay Health Code | China | Centralized [26] | GPS [26] | This app is using individual QR codes, phone location, facial recognition, and drones to identify an individual's movement, this is threatening individual privacy. The Alipay app encrypts the storage, and it provides the chance to the individual to request personal data deletion. |
| Korona Önlem App | Turkey | NA | NA | The app requests access to contacts microphone, camera and calendar. Does not clearly if data is encrypted or not and if the users are able to request deleting their data. |
| Kwarantanna Domowa | Poland | NA | NA | The user is able to view the collected data about him following the EU's General Data Protection Regulation (GDPR) that appears unencrypted and the data will be stored for six years after the user deactivated his account. |
| Location-Based Text Alerts | South Korea | NA | GPS | Infected users' data are only collected |
| Moscow's patient-tracking app | Russia | Centralized | GPS | Collects users' location data if the user provides access in Moscow. |

(Continued)

| Application name | Country | Model | Technology | Privacy concerns |
|---|---|---|---|---|
| North Dakota Care19 App | U.S.A | Decentralized | GPS | This app gives the user a random ID and stores the location data only without personal data, the location will be tracked during the day, and the data deletion will be after 14 days. |
| PACT | U.S.A | Decentralized | Bluetooth | Only the Bluetooth tokens of infected users are sent to the authority and the authority will create and send to the infected user a permission number that will be announced publicly associated with all the contacts event numbers, then the public will be able to compare their contact event number with the announced number to know if they get affected. |
| Pan-European Privacy-Preserving Proximity Tracing (PEPP-PT) | EU | Decentralized | Bluetooth | Only the Bluetooth tokens of infected users are sent to the authority and the authority will create and send to the infected user a permission number that will be announced publicly associated with all the contact event numbers, then the public will be able to compare their contact event number with the announced number to know if they get affected. |
| PretoGO | Poland | Centralized | GPS based | Papers did not discuss its privacy violence clearly but for sure it has been mentioned as one of the applications that has privacy problems. |
| Privacy-Preserving | UK | Centralized | Bluetooth | They clearly state their privacy policies and what the data will be stored, infected users' identity is anonymized after they agreed to transmit their data to the authority. Whereas the uninfected users' data will stay local. |
| Rakning C-19/Tracing C-19 | Iceland | Centralized | GPS | infected users' identity is anonymized after they agreed to transmit their data to the authority. Whereas the uninfected users' data will stay local. Both data are encrypted either locally stored or on the central server which will be deleted after 14 days. |
| Safe paths | U.S.A | Decentralized | GPS | The uninfected users' information will be locally stored but if they get infected their location will be encrypted and shared with SafePlaces web application, then healthy users will use PrivateKit application to compare their personal location with the announced locations of the infected persons. The user has the ability to delete the local data but not the one that has been shared with SafePlaces. |
| Self-Quarantine Safety Protection | South korea | Centralized | GPS | The data are collected from many locations, CCTV, and credit card usage and sent to Korea Centers for Disease Control (KCDC) twice a day. Only the user's locations are reported immediately to KCDC to ensure that the user does not leave his quarantine, which causes high stress to the users. The infected users' data are deleted after 2 months and all users' data will be deleted after 6 months if they deactivate their accounts. |

| Application name | Country | Model | Technology | Privacy concerns |
|---|---|---|---|---|
| Shin Bet Mobile Data Monitoring | Israel | NA | GPS | The data is not encrypted, and it will be deleted after 6 months if they did not find the need to extend the time. |
| Smittestopp App | Norway | Centralized | GPS+ Bluetooth | The app collects users' phone number, location, age, mobile operating system and phone model. All these data will be deleted after 30 days. |
| StayHomeSafe | Hong Kong | Centralized | GPS | Collects users' location data by scanning their wristband's QR code to track the users if they left their home and force the passengers to do so. |
| StopCovid | France | Centralized | Bluetooth | This app was developed based on the TraceTogether model, the France government adopted Bluetooth technology rather than GPS to protect the privacy still Bluetooth technology is vulnerable to data breaches. Until now the anonymity level is preserved but it is expected to be sacrificed if the number of users increased. |
| Stopp COVID-19 | Australia | Centralized | Bluetooth technology | Send data to the government central server. |
| Swiss Covid | Switzerland | NA | Bluetooth (Google/Apple API) | This app faces privacy concerns upon the third-party API's used since it may reach the data, another concern is that this app does not provide control over data. |
| Tabaud | Saudi Arabia | NA | Bluetooth | They are conflict on using innovation or protect users' privacy. |
| trace and track | UK | Centralized | Bluetooth | This app provides an anonymity level since the ID is anonymous and data is stored on the phone then after 14 days it will be uploaded to the cloud. |
| TraceTogether | Singapore | Centralized | Bluetooth (BlueTrace protocol) | It does not store any geolocation or personal data but requires phone numbers that are anonymous for the public but not anonymous for the government, which raises the concern about government privacy protection policies. First, there is a security risk since the data is not encrypted and it is vulnerable to malicious attacks. Second, there is an ethical risk for the infected people because if an individual passes by an infected person there will be an alert to notify the healthy person. However, data is stored locally on the individual's phone and after 21 days it will be deleted. |
| Tracking App | South Korea | Centralized | NA | Privacy issue since the government controls the data. |
| Virtuous Surveillance | South Korea | Centralized | GPS | This app several privacy concerns, as it publicly announces the infected user's information that include: last name, gender, credit card history and all recent location visits. |

approaches to protecting users' privacy and the anonymity degree. In the centralized model, health authorities and governments collect data from individuals regardless of whether they are healthy or diagnosed with COVID-19, mapping the collected information to everyone uniquely in a central server. This approach effectively controls cases if it is

widely used, as it provides a comprehensive view. However, the model does not ensure users' privacy because there is no control over data sharing. In contrast, the decentralized model does not offer public control, as it does not have a central server for data storage. Instead, individuals who tested negative or did not test at all store their data locally on their devices and can check whether they were in touch with infected people through public platforms that have already gone through a data anonymization cycle. The sections below discuss the models in detail (*Wang & Liu, 2020*).

## Centralized contact-tracing

As abovementioned, there are two different technologies to collect individuals' information used in centralized and decentralized models. This section will discuss the differences between using GPS and Bluetooth technologies in the centralized model from a privacy perspective. In centralized GPS-based applications, the user's places are collected and shared with the authorities' servers (*Joo & Shin, 2020*). In centralized Bluetooth-based applications, data are collected by creating random tokens at different times and exchanging with other users if they happen within 6 feet. Later, each user's phone number and token are sent to health authorities, informing the people that person has countered within the past 2 weeks if the user is found to be infected. A centralized application using both GPS and Bluetooth technologies to collect users' information has been deployed in India (*Wang & Liu, 2020*).

The main examples of centralized contact-tracing applications are Alipay Health Code used in China and Self-Quarantine Safety Protection used in South Korea (*Joo & Shin, 2020*), both of which are GPS-based, and TraceTogether, a Bluetooth-based app used in Singapore (*Wang & Liu, 2020*). These apps have helped governments limit the spread of COVID-19. In the centralized model, control is vested in governments and authorities, since they can trace all users' health status and the number of infected people; this makes such apps more accurate and efficient in quickly understanding the situation. Thus, they allow for better control of the virus (*Riemer et al., 2020*).

When comparing Asian countries with some European countries to adopt centralized contact-tracing apps, Asia have shown better control over the virus' spread than Europe. The reason can be attributed to Asian citizens' willingness to sacrifice privacy in the interest of public health (*Cha, 2020*), whereas in some European countries, such as France with the Stop-COVID app, citizens refused to use contact-tracing, triggering a massive virus spread and losing control (*Rowe, 2020*). In other words, using the most effective approach does not guarantee the best results because civic readiness for and app's acceptance are crucial influencing factors.

Centralized applications have shown great potential to limit viral spread; however, users' concerns about privacy and how their data are exposed and controlled by government and health authorities have caused stress and motivated them to avoid using contact-tracing apps. This stress is known as technostress: technology-caused anxiety and negative emotions. The Alipay Health Code used in China is a great example of how collecting users' information increases stress and anxiety. Considering its different information collection methods, like drones, GPS, QR codes, and CCTV (*Joo & Shin,*

*2020*), it can be agreed it is difficult to trust the government and health authorities when they do not state how they protect and process the collected information (*Vitak & Zimmer, 2020*; *Nabity-Grover, Cheung & Thatcher, 2020*).

Moreover, using centralized techniques has raised some privacy risks associated with each technology. For GPS-based applications, the authority collects information from all users regardless of infection status and broadcasts all the locations the user has visited recently when someone tests positive, making it hard to maintain infected users' confidentiality. Bluetooth-based applications are also vulnerable to these stated risks. Besides, there is a risk the authorities will connect with another database *via* users' phone numbers and access sensitive information if deemed necessary (*Wang & Liu, 2020*). Furthermore, some people think the centralized contact-tracing approach infringes freedom and carries long-term risks like records of the information collected during the pandemic even after ending (*Rowe, 2020*).

However, one study suggests forcing citizens to use contact-tracing apps regardless of privacy concerns applying a framework complete with rules and policies punishing non-users will enhance control and limit the virus' spread (*Riemer et al., 2020*).

## Decentralized contact-tracing

Based on the aforementioned problems with centralized contact-tracing apps, the importance of using decentralized platforms offering a higher degree of privacy regarding individuals' data has emerged (*Cho, Ippolito & Yu, 2020*). Decentralized contact-tracing provides more privacy because, unlike in the centralized model, it has mechanisms to verify privacy and use public and private keys and digital signatures (*Skoll, Miller & Saxon, 2020*)

The solutions provided in the distributed approach can utilize either GPS or Bluetooth, both offering a degree of privacy. With Bluetooth, data are transferred in a phone-to-phone transaction: location data are sent directly without an HTTP connection. Compared to GPS, Bluetooth is more private, as GPS applications share data *via* HTTP protocol. However, both can offer more privacy when used with blockchain (*Garg et al., 2020*).

BayesCOVID is a GPS approach combining contact-tracing, symptom tracking, and Bayesian network. This approach gives the user a choice, since it provides a user contract that enhances the application's utility (*McLachlan et al., 2020*).

One important solution is the Apple/Google Bluetooth approach that prioritized individuals' privacy by using Privacy-Pre-Serving Proximity Tracing (DP-3 T) with both the SARS and the Middle East Respiratory Syndrome (MERS) outbreaks. This method provides a high degree of privacy, since it uses Bluetooth, which offers more location privacy than GPS (*Fahey & Hino, 2020*; *Wang & Liu, 2020*) and avoids storing data (location, users' identities, contacts) in archives. This approach is very secure and provides a high degree of privacy. Unfortunately, it taxes on smartphones' battery life, which is a drawback for users (*Fahey & Hino, 2020*).

However, offering users control is encouraging. Specifically, users can control which data are being transferred and which are not. This approach uses two models: simple data

transfer and distributed computation. It also ensures self-awareness, which will encourage users to risk sharing data (*Nanni et al., 2020*).

Furthermore, the decentralized technique provides a high degree of privacy because it uses two-factor authentication blockchain. It also uses personal data without storing it in a database. It accomplishes this with encoded data storage, which can be accessed with users' consent. This approach cryptographically signs user data and stores it. When the data are deleted, the hash will redirect to a null reference called "orphan hash." This approach has the advantage of encouraging users to trust contact-tracing applications, which will help restore normal life sooner (*Eisenstadt et al., 2020*).

Based on the outcome of this SLR and our findings, we present the following future considerations and directions for contact tracing apps and related technologies in the fight against COVID-19 and future pandemic outbreaks that are worth investigating and implementing to encourage adoption by the wider population:

Utilizing privacy-protecting technologies such as Artificial Intelligence (AI) and Machine Learning (ML) is suggested to help analyze the level of infection by viruses through identification of infected areas, tracing, and monitoring infected people (*Vaishya et al., 2020*; *Ahmed et al., 2020*). Other researchers (*Yang et al., 2020*; *Ting et al., 2020*) propose the use of the Internet of Things (IoT) and thermal imaging devices (*Chamberlain et al., 2020*; *Mohammed et al., 2020*) to track positive cases and control the wide spread of COVID-19 virus. Additionally, some studies proposed use of a privacy-preserving contact-tracing scheme through blockchain-based medical applications (*Zhang et al., 2021*; *Chang & Park, 2020*).

Governments, decision makers, and public health authorities must implement a proper feedback system throughout contact tracing apps deployment phases to gain public trust and increase adoption levels. It must be very clear to users what data is being collected, who is accessing the data, and how it is being used. It is key to study and understand human behavior throughout the design and development phases of the apps before their actual implementation. Authorities could implement multiple models and theories, such as the technology acceptance model, diffusion of innovation model, and motivation theory, to study the acceptance and usage level of future contact tracing technologies (*Lucivero et al., 2020*; *He, Zhang & Li, 2021*; *Sharma et al., 2020*).

Lastly, it is necessary to focus on privacy and data transformation while minimizing data collection and access to reduce contact-tracing privacy concerns (*Fahey & Hino, 2020*).

## Privacy protection laws for COVID-19 applications

Data privacy, also known as information privacy, is a subset of data security that focuses on data management while complying with data security guidelines. The essence of data privacy is how data should be collected, stored, managed, and shared. Practical data privacy issues often revolve around whether or not data is shared with third parties, how it is shared, and how data is lawfully collected and preserved. With the rise of the digital economy, one of the most difficult issues for organizations to address is data privacy. As a

result, adhering to a data privacy policy and managing the data that is required is crucial in order to gain users' confidence.

With the individual as the major character, data privacy involves not only the proper handling of data but also the public expectations about privacy. Individuals are entitled to privacy and control over their personal information. Procedures for safely and securely keeping, processing, acquiring, and sharing personal data must be implemented at all times.

Consumers' data protection and privacy are vital in today's technology era, therefore governments, healthcare providers, and business groups have been using digital tracking to keep COVID-19 outbreaks under control. Although this method has the potential to minimize pandemic transmission, it has significant privacy implications (*Sharma et al., 2020*; *Monroe, Tazi & Das, 2021*).

There is tension between privacy and information disclosure, and the data's privacy managed by digital health apps must be maintained to limit the virus spread. There are two types of COVID-19 data. The first type consists of cases and special medical data, such as disease statistics, medical sources, and the history of cases in contact with the disease. The second type is data related to government-imposed containment policies and health measures, such as social distancing and quarantine.

A problem has emerged in the data managed in COVID-19 applications and on social media: data and information are released from various sources, confusing the public about what information is true and which sources are credible. The main sources of COVID-19 data and information are government agencies, local governments, health authorities, and international organizations such as the World Health Organization (WHO). The data and information administered by COVID-19 digital health apps has helped limit the epidemic spread through disclosure, specifically facilitating people's understanding of the containment measures. This has been effective because people are more willing to comply with containment measures when they understand the issues raised around the virus. Additionally, this will result in raising awareness and acceptance of policies, and an enhanced sense of safety.

Finally, information disclosure reshapes people's perceptions of the epidemic and containment measures, enabling the public to overcome difficulties and limit the virus' spread. However, disclosure should be performed only with individual consent (*Fu, Ma & Wu, 2020*).

### Protection law

COVID-19 is the most recent danger posing a threat to the world's health and economic sectors. Tracing the main and secondary contacts of confirmed COVID-19 cases using contact-tracing technologies and devices is one of the most effective approaches to reduce the spread of the virus. The European Union (EU) emphasized the importance of data protection and privacy in digital measures, stating that data must be used exclusively for the intended purpose, that is, to prevent the spread of disease. Strategies to contain the pandemic include the use of technology to contain and warn those who have been in contact with people infected.

The benefits gained by tracking people are greater than the potential of losing users privacy because of its desired benefits to eliminate virus outbreaks (*Galloway, 2020*; *Schneble, Elger & Shaw, 2020*). The EU, which has a robust data protection system, requires that all states share personal data that has been collected through contact-tracing applications.

The US Government and the public sought to develop consumer data privacy protection laws. The value of privacy was stressed by multiple entities including elected leaders, members of congress, and others by emphasizing a myriad of possible harms connected with its violation. Privacy protection also helps guard against fraudulent or economic damage caused by identity theft, fraud, extortion, or other acts of crime. Guaranteeing information privacy is also important to reduce public fears of divulging details of their private lives, such as their personal contacts or behavioral habits, in the context of health-related data.

Most mobile phone applications that track symptoms or trace contacts require widespread use among the population, which is only achieved if users trust these apps. Some of the common privacy protection criteria of smartphone apps are transparency, purpose, anonymity, informed consent, time limits, and data management.

Transparency requires straightforward software reporting policies, which help to create public confidence in governments and health organizations, all of which are important to promote informed and voluntary use of COVID-19 related services.

Anonymization is the use of a series of mechanisms to prevent data from being associated to a specific person. Informed consent ensures that consumers have the information they need to make informed decisions about willingly releasing confidential personal data in order to respond to public health requirements. Time limits means ensuring that the data obtained about contacts, location tracking, and mobile device proximity can only be used in the scope of the crisis. Data management means applying all protections measures throughout the life cycle of data-collection systems for contact-tracing mobile apps (*Boudreaux et al., 2020*).

COVID-19 apps should be free from security and privacy problems because these aspects are important to users. Legal protection must also be provided because a lack of privacy will lead to application failure if users shy away from using these applications due to a lack of trust (*Hendl, Chung & Wild, 2020*; *Culnane, Leins & Rubinstein, 2020*; *Islam et al., 2020*).

### Ethical

In addition to privacy concerns, there are many ethical issues related to the data collection processes and algorithms of contact-tracing apps. During the COVID-19 pandemic, governments and healthcare organizations rely on location and health data to assess infection rates, effectiveness of social distancing measures, and disease transmission rates. As COVID-19 spreads, several COVID-19 tracking applications were developed to aid in the containment of the pandemic. A framework has been established to validate COVID-19 apps' ethics. It is intended to assist designers and publishers of contact-tracing apps in determining the application's ethical justification. If used properly, the apps should

be a major component of disease management, proportional to the severity of the public health threat and scientifically sound and time-bound ethical design. Following the COVID-19 breakout, these groups must utilize this data in an ethical, robust, and transparent manner to prevent widespread skepticism and any breaches of privacy rights (*Klar & Lanzerath, 2020*).

### Values

When viruses start to cause safety issues, the balance between the need to fight the virus and the obligation to uphold individual rights often changes. COVID-19 tracing apps illustrates how close monitoring and contact-tracing compromises privacy rights around the world. In response to the epidemic, the Australian government created COVID-19 Safe, a mobile phone app for contact-tracing. Governments around the world have also been using technology to help maintain social distancing, isolation, and contact-tracing.

Moreover, with privacy as a major concern, COVID-19 apps must be built explicitly based on values including fairness, equality, solidarity, and user benefit. Due to the high privacy impact, data sharing is only possible when there are serious health conditions. There are cases in which governments receiving personal data must be identified. All population groups should be able to use COVID-19 surveillance technology which respects their privacy (*Lodders & Paterson, 2020*; *Van Kolfschooten & de Ruijter, 2020*; *Lee & Lee, 2020*). The use of the contact-tracing process is unprecedented and could have serious consequences for public health. It is necessary to implement public-interest digital technology practices that are in line with values (*Lodders & Paterson, 2020*; *Van Kolfschooten & de Ruijter, 2020*; *Lee & Lee, 2020*).

### Issues

As mentioned before, employing digital surveillance technologies to contain the virus raised several privacy issues related to the use, storage, and manipulation of collected personal information. Individuals' concerns about their privacy prevented them from using such apps, which obviously affected the tracing process considerably. Those individuals questioned the integrity of the data collection and utilization processes and whether the data will be anonymous, temporarily stored, or open to public use. Figure 6 illustrates the most important privacy issues of contact-tracing apps.

To fulfill privacy guidelines to the highest degree, tracing apps should clearly reveal for which purpose the data will be used, and whom will have access to it and control it. In addition, the data should only be stored by authorized agencies. Health care agencies must clarify what will happened to the data in the future after the pandemic is over. All tracing apps must comply with the international privacy standards to minimize public privacy concerns.

On the other side, these issues can be eliminated from technological perspective by applying more secure and private models. Two popular models have been used in designing tracing apps, which are centralized and decentralized. After comparing these two models, the decentralized model proved to be more secure and reliable, especially when used with Bluetooth technology.

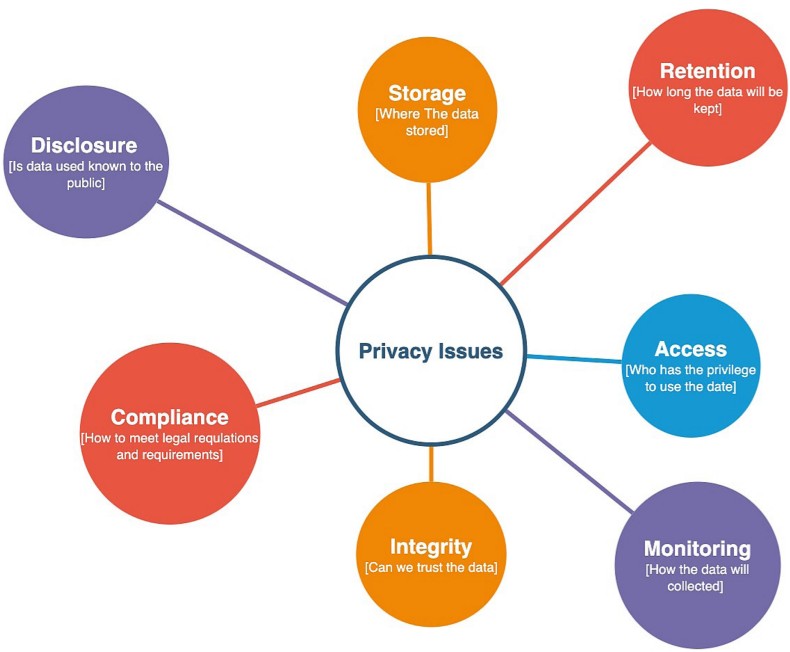

**Figure 6** **Privacy issues of contact-tracing applications.**

### Entities contributing to preserving privacy of healthcare applications

In 1996, a law obliged the health authorities to protect patients' privacy, disclosing their data only in high-risk cases and guaranteeing information privacy even during data exchange. Furthermore, integration between citizens' data and the health authorities will help identify infected patients and restrict their interaction with healthy people. Patients' privacy is one of the most important topics raised during the COVID-19 pandemic.

COVID-19's proliferation has created unique circumstances that have prompted a change toward telemedicine infrastructure adoption. Telemedicine has become an important part of clinical care delivery, and many medical institutions report a significant increase in the use of telemedicine. For example, One Medical Center in New York City, witnessed a major increase in urgent care virtual visits from 102 per day pre-COVID-19, to 802 per day post-COVID-19. As the transition to telemedicine progresses, new problems and dangers have emerged, especially in the areas of information security and privacy.

The Privacy Rule, issued in December 2000 by the US Department of Health and Human Services (HHS), protects the privacy of individually identifiable health information. In addition, The European Union, passed a data privacy legislation to protect patient's personal health information, and has one of the most effective data protection and privacy policies in the world. However, government agencies around the world have warned that the risk of cyberattacks against healthcare departments and institutions researching COVID-19 is increasing since the pandemic started (*Jalali, Landman & Gordon, 2021*).

There are many applied methodologies when it comes to the type of data health officials are sharing with the public. In the US, the local government in Los Angeles County provides an estimated age distribution of patients, and a breakdown of the number of cases

in more than 140 cities and communities. However, residents in Florida are given much more information, including the cities affected, the number of people tested, the age distribution of cases, and the number of cases in nursing homes.

In response to the COVID-19 pandemic, the Indian Ministry of Health and Family Welfare issued guidelines for the mandatory notification of information for COVID-19 patients, allowing the government to enact any regulations it deems necessary to prevent the outbreak or spread of such epidemics. The Indian government requires doctors to report COVID-19 cases and suspected cases to designated government agencies, and the government agencies can then respond appropriately to limit the disease spread based on the information provided by the health care professionals (*Fu, Ma & Wu, 2020*; *Shekhawat et al., 2020*).

Digital health apps are available in Apple Store and Google Play App Store that contain more than 318,000 available applications which are updated daily, to comply with the latest policies announced by the health authorities.

The Federal Data Protection Act (FADP) provides a comprehensive framework dealing with data protection using defined principles. It insists on securing individuals' privacy and provides protection measures. According to FADP, patient data are considered sensitive and require additional privacy. Each user has to give consent for the health authorities to use their data for health purposes. Therefore, the applications have to grant the user the right to revoke, update, and remove the data (*Vokinger et al., 2020*).

The COVID-19 pandemic has shed light on digital health applications but ignored their data privacy. Blockchain technology appears to be an ideal solution to secure and authenticate certificates, health and medical records, and prescriptions, while preserving privacy (*De, Pandey & Pal, 2020*) Doctors may disclose patients' information to competent authorities under specific circumstances for society's greater interest. Quarantine and social isolation measures imposed by the health authorities have effectively limited viral spread. Thus, it is important to collect individuals' information *via* contact-tracing apps (*Labs & Terry, 2020*; *Shekhawat et al., 2020*).

In conclusion, tracing applications help to control the spread of the COVID-19 pandemic. Governments and health authorities developed these apps based on many technologies and different models. The aim of these apps is to monitor the infected individuals and keep track of the public status. Some countries had followed international laws and local regulations to protect users' privacy while designing and developing these apps. However, other countries did not comply with these requirements, which resulted in privacy breaches. Nevertheless, there was no clear guidelines regarding the disclosure of using the personal data in tracing apps. However, even with the privacy limitation in tracing apps, WHO stated that they had effectively helped in containing the pandemic and slowing down its spread.

COVID-19 has put forward privacy concerns in many fields, and this systematic literature review has discussed it from three perspectives: the public's privacy in using contact-tracing apps, laws and policies that should be followed to protect users' privacy and digital authorities' strategies for dealing with data privacy. The classifications of the included literature is illustrated in Fig. 7.

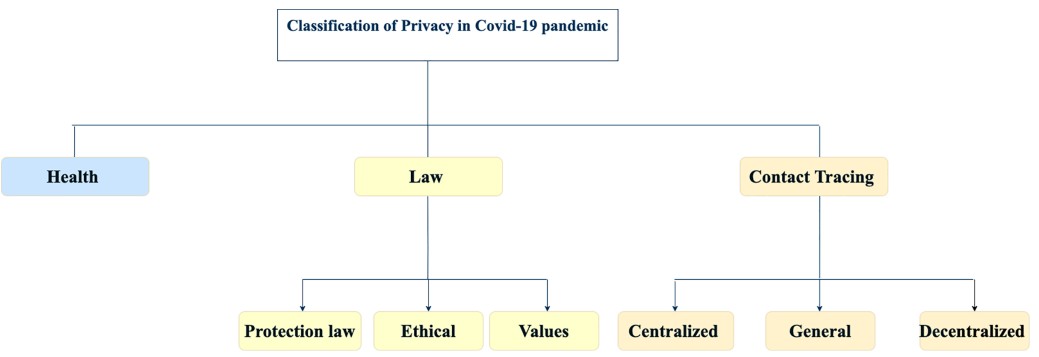

**Figure 7 Classification of privacy during the the COVID-19 pandemic.**

## Implications

Many persons avoid using COVID-19 apps due to privacy concerns about location and health data (*De, Pandey & Pal, 2020*; *Fahey & Hino, 2020*). To prevent this issue, governments should develop policies ensuring individual data privacy rights, encouraging people to trust these apps and provide their information voluntarily (*Lee & Lee, 2020*).

Another implication: not all citizens have Internet access. The solution is providing universal Internet so that everyone can access COVID-19 apps. Moreover, some individuals do not have their own devices, raising the issue these apps may not cover all citizens. This challenge can be surmounted by assigning one account per family to reach the highest number of users (*De, Pandey & Pal, 2020*; *Rowe, 2020*).

Lastly, COVID-19 problems are widespread since it is a new virus, but researchers are trying to address each concern to reduce the virus' effects and minimize its spread (*De, Pandey & Pal, 2020*; *Hendl, Chung & Wild, 2020*)

## Limitations and future work

COVID-19 has revealed many limitations in many areas such as information management and data privacy; therefore, digital surveillance has become more important than ever. Artificial intelligence, big data, the Internet of Things, and GPS have been recognized as paramount technologies in developing COVID-19 contact-tracing apps (*Mbunge, 2020*; *Joo & Shin, 2020*; *Fahey & Hino, 2020*).

Privacy protection is an important issue. In the context of digital health and the COVID-19 epidemic, within a framework for evaluating applications from epidemiological and legal perspectives, solutions are designed to obtain useful information with several limitations, including preventing sharing sensitive personal information (*Sharma et al., 2020*; *Nanni et al., 2020*; *Vokinger et al., 2020*).

The studies did not cover all classifications of COVID-19 research's keywords, and they are also restricted to specific countries' cultures and policies. Moreover, the studies did not consider the differences in values and cultural and political aspects of the countries using tracking applications. Additionally, no study has discussed contact-tracing applications developed after April 30, 2020 (*Garg et al., 2020*; *Trang et al., 2020*; *Islam et al., 2020*;

*Kumar, Shahrabani & Das, 2020*). For the future, Garg expects to develop an RFID solution and aims to reduce the cost of scaling the RFID range (*Garg et al., 2020*).

The COVID-19 pandemic is a recent one, hence applications in this field are limited. The study found that a limited number of mobile applications were developed, which will be used as a benchmark for future applications' specifications and learn more about users' interactions on different national app platforms. That result makes a significant contribution to health institutes and health practitioners (*Trang et al., 2020*; *Islam et al., 2020*).

During epidemics, it has been suggested it is prudent to allow for some loss of privacy and place trust in smart technologies to help fight deadly, invisible creatures. With the continuing spread of COVID-19, Singapore will continue to deploy technological tools and interventions (*Lee & Lee, 2020*).

A limitation of future research on an ultimate dependent variable is the adoption of COVID-19 applications recommendations for pre- and post-testing in future studies. Emphasis should be placed on collecting data about infectious diseases, ensuring public health and that epidemiological surveillance technology features are ethical and reflective of fair values, and reducing the vulnerability of at-risk individuals (*Sharma et al., 2020*; *Skoll, Miller & Saxon, 2020*; *Whaiduzzaman et al., 2020*; *Hendl, Chung & Wild, 2020*).

Moreover, developing and improving the efficiency and effectiveness of information systems and technology in organizations as well as monitoring people's safety and privacy in the fight against COVID-19 (*Mbunge, 2020*; *O'Leary, 2020*; *Wang & Liu, 2020*) are essential. Additionally, expanding anti-snooper privacy safeguards, imposing usage restrictions in contact-tracing, and adding a private messaging system will enhance overall privacy. There have been discussions about creating an application to track contacts directly using Bluetooth (*Cho, Ippolito & Yu, 2020*; *Vitak & Zimmer, 2020*).

## CONCLUSION

The effect of the COVID-19 pandemic represents an enormous challenge to public health authorities and governments around the world. The pandemic put major pressure on health systems and resulted in fundamental changes to everyday life for individuals and organizations. Public health authorities introduced contact-tracing systems which include the use of digital contact-tracing mobile apps. Contact-tracing apps are promising technologies for rapid tracing and tracking of infected persons, and they can support manual contact-tracing and tracking methods to control the COVID-19 virus. However, some people avoid using digital surveillance apps altogether since they are concerned about their privacy. Governments and health authorities should address this issue and try to preserve the rights of those who do not wish to waive their privacy.

In most countries, the use of these apps is not mandatory, which makes it challenging to predict their acceptance and participation levels. It is significant for governments and health officials to gain the trust of their citizens and show suitable transparency by clarifying what personal data is collected and how it is being used. The efficiency of contact-tracing apps is highly dependent on how authorities address all related privacy challenges and concerns. Their efforts will surely determine the role of digital contact-

tracing technologies in future pandemic occurrences and lessons learned from similar errors.

The challenges facing contact-tracing apps include, in addition to privacy, technical, usability, and addressing additional requirements reported by some users. A considerable number of contact-tracing apps were not welcomed by the public and suffered low acceptance levels, which dramatically affected their efficiency. As an example, only the Singaporean app had a penetration level of a little over 30%, the Australian and Swiss apps had a penetration level below 20%, and the penetration values for the majority of other apps around the world were less than 5%.

The volume of personal data contact-tracing apps collected varied considerably, some apps collected absolutely no data while others collected a significant amount of highly private personal data. The majority of the surveyed apps did not give users an option to deactivate the app, such as logging out, without uninstalling them. Additionally, the lack of standardization for contact-tracing technologies resulted in fragmented non-interoperable apps. As countries are coming out of lockdown and reopening borders, there is an essential need for a unified and interoperable contact-tracing app that can easily be implemented globally without compromising users' privacy.

A possible solution to the privacy issues and concerns can be implemented through a comprehensive government-mandated data privacy policy in the context of digital health applications. Another option is for governments to deploy fully decentralized and highly accurate applications, which do not keep any records of sensitive personal information and provide the same level of accuracy as the centralized approach. One suggestion for a decentralized approach is to use a blockchain-based app algorithm balancing users' privacy and public health requirements. Moreover, Internet intermediaries must work with governments and civil society to address privacy and surveillance issues to improve new contact-tracing technology adoption levels in the future.

## ACKNOWLEDGEMENTS

We thank King Abdulaziz City for Science and Technology (KACST) and King Saud University (KSU) for providing all the necessary technical tools for data collection and analysis.

### Funding

The authors received no funding for this work.

### Competing Interests

The authors declare that they have no competing interests.

### Author Contributions

- Amany Alshawi conceived and designed the experiments, performed the computation work, prepared figures and/or tables, and approved the final draft.

- Muna Al-Razgan conceived and designed the experiments, performed the computation work, prepared figures and/or tables, and approved the final draft.
- Fatima H. AlKallas performed the experiments, prepared figures and/or tables, and approved the final draft.
- Raghad Abdullah Bin Suhaim performed the experiments, authored or reviewed drafts of the paper, and approved the final draft.
- Reem Al-Tamimi analyzed the data, authored or reviewed drafts of the paper, and approved the final draft.
- Norah Alharbi analyzed the data, authored or reviewed drafts of the paper, and approved the final draft.
- Sarah Omar AlSaif performed the experiments, authored or reviewed drafts of the paper, and approved the final draft.

## Data Availability

This is a literature review.

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
