# Peer review of "Data privacy during pandemics: a systematic literature review of COVID-19 smartphone applications"

_PeerJ Computer Science, doi:10.7717/peerj-cs.826_

## Round 0.1 · original submission · Major Revisions

According to the reviewers' comments, the paper needs improvements before publishing. Recent and important papers should be discussed. Conclusions should be extended. Sections are not given in detail.

·

Basic reporting

This literature review has followed a comprehensive methodology to study different digital technologies such as CTAs which have been used to control the COVID-19 spread. The manuscript has been written clearly using professional English which is easy to understand.

Experimental design

The article content has fallen in the Aims and Scope of PeerJ Computer Science.

The study has done a thorough investigation with a high ethical standard on all kinds of contact tracing applications (CTAs) and reviewed 800+ papers published in some prestigious journals such as Science Direct, IEEE, Scopus, etc.

Validity of the findings

Conclusions are well stated, and valid. The result and recommendations made are beneficial to governments and countries all over the world.

Additional comments

Figure 1 is a bit ambiguous. I would suggest changing it as in the attachment.

Reviewer 2 ·

Basic reporting

- The paper is very well written and in simple words. It is thus easy to understand which is a very important characteristic of any document.

- The sections are titled as questions. While this is innovative and gets the point across, I prefer a more conventional approach to naming sections

- Each section and each point has to be discussed in much more detail. The authors superficially discuss the ideas of various works and do so in a rather heterogeneous manner with separate paragraphs dedicated to separate papers. I would look forward to a more homogeneous discussion of ideas with disparate work smoothly falling into the discussion.

Experimental design

- Very little discussion is dedicated to the users’ privacy and its breach. The authors discuss laws that are prevalent around the world, “ethical” issues, and “values”. I would prefer a much more detailed analysis of the issues around privacy and its breach first in a general context and subsequently specifically with respect to the pandemic and the tracing apps.

- The authors discuss the attempts made by health authorities to protect and preserve privacy. I feel this discussion is limited. The authors should also include the efforts made by governments, health authorities, world bodies, and also those made at individual levels.

Validity of the findings

- Conclusions on various aspects of the study are not drawn. The authors talk about the issues based on the contributions of various endeavours but do not draw appropriate conclusions based on these.

Additional comments

- A few figures, block diagrams etc. would help the reader more effectively grasp the ideas being discussed.

- The authors may want to justify the content and both ends for a better appearance.

---

## Round 0.2 · accepted · Accept

The comments have been addressed. We are pleased to inform you that your manuscript has been accepted for publication after the language corrections.

·

Basic reporting

No coment

Experimental design

No comment

Validity of the findings

No comment

Reviewer 2 ·

Basic reporting

My concerns on basic reporting have been addressed.

Experimental design

My concerns regarding the study design have been addressed.

Validity of the findings

My concerns regarding the validity of the findings have been addressed.

Additional comments

All my concerns have been addressed. I am happy to recommend that the paper be accepted.